# Dietary Outcomes of the ‘Healthy Youngsters, Healthy Dads’ Randomised Controlled Trial

**DOI:** 10.3390/nu13103306

**Published:** 2021-09-22

**Authors:** Lee M. Ashton, Philip J. Morgan, Jacqueline A. Grounds, Myles D. Young, Anna T. Rayward, Alyce T. Barnes, Emma R. Pollock, Stevie-Lee Kennedy, Kristen L. Saunders, Clare E. Collins

**Affiliations:** 1School of Health Sciences, College of Health, Medicine and Wellbeing, Priority Research Centre for Physical Activity and Nutrition, University of Newcastle, Callaghan, NSW 2308, Australia; lee.ashton@newcastle.edu.au (L.M.A.); clare.collins@newcastle.edu.au (C.E.C.); 2School of Education, College of Human and Social Futures, Priority Research Centre for Physical Activity and Nutrition, University of Newcastle, Callaghan, NSW 2308, Australia; jackie.grounds@newcastle.edu.au (J.A.G.); anna.rayward@newcastle.edu.au (A.T.R.); alyce.barnes@newcastle.edu.au (A.T.B.); emma.r.pollock@newcastle.edu.au (E.R.P.); stevielee.kennedy@newcastle.edu.au (S.-L.K.); Kristen.saunders@newcastle.edu.au (K.L.S.); 3School of Psychology, College of Engineering, Science and Environment, Priority Research Centre for Physical Activity and Nutrition, University of Newcastle, Callaghan, NSW 2308, Australia; myles.young@newcastle.edu.au; 4Hunter Medical Research Institute, New Lambton Heights, NSW 2305, Australia

**Keywords:** preschool-aged children, dietary intake, parenting, fathers, intervention

## Abstract

(1) Background: The effect of fathers on dietary intake in preschool-aged children is under-explored. The aims were to: (i) evaluate the efficacy of a family-based lifestyle intervention, *Healthy Youngsters, Healthy Dads*, on change in dietary intake in fathers and their preschool-aged children post-intervention (10 weeks) and at 9 months follow-up compared to a waitlist control group and (ii) investigate associations in father–child dietary intakes. (2) Methods: Linear mixed models estimated group-by-time effects for all dietary outcomes, measured by food frequency questionnaires. Cohen’s *d* determined effect sizes, while correlation tests determined associations in father–child dietary intakes. (3) Results: For children, medium group-by-time effects sizes were identified at 10 weeks for sodium intake (*d* = 0.38) and percentage energy from core foods (*d* = 0.43), energy-dense, nutrient-poor (EDNP) foods (*d* = 0.43) and prepacked snacks (*d* = 0.45). These findings were sustained at 9 months follow-up. For fathers, medium to large, group-by-time effect sizes were identified at 10 weeks for energy intake (*d* = 0.55), sodium intake (*d* = 0.64) and percentage energy from core foods (*d* = 0.49), EDNP foods (*d* = 0.49), and confectionary (*d* = 0.36). For all of these dietary variables, except sodium, effects were sustained at 9 months. Moderate to strong associations existed in father–child dietary intakes for some of the dietary variables. (4) Conclusions: Although further research is required, this study provides preliminary support for targeting fathers as agents of change to improve dietary intakes in their preschool-aged children.

## 1. Introduction

Improving diet quality of young children represents a major public health challenge at both national and international levels. Large-scale nutrition data across several countries have reported low fruit and vegetable intake [1,2], high sodium intakes [3], poor diet quality [4,5,6], and high consumption of energy-dense, nutrient-poor (EDNP) foods [3,7] in preschool-aged children. Early childhood presents a crucial time to implement positive diet patterns because dietary habits formed early in life can influence eating habits across the lifespan [8]. Furthermore, engagement in healthy eating practices in the early years is associated with favourable health and developmental outcomes, such as lower adiposity [9], better lean mass [10], psychosocial health [11] and enhanced cognitive abilities [12].

The diet quality of men is also a concern. Globally, poor diet quality is the third leading risk factor, accounting for 4.5 million (3.7–5.5) deaths, or 14.6% (12.0–17.6) of all male deaths in 2019 [13]. A global analysis from 187 countries found that males had significantly worse dietary patterns than females (*p* < 0.0001) [14]. This indicates that they have lower intakes of nutrient-rich foods such as fruits and vegetables, and wholegrains, and higher intakes of EDNP foods such as processed meats and sugar-sweetened beverages (SSBs). Furthermore, this global analysis also indicates that men within the age groups of 20–29 years and 30–39 years have the worst diet quality compared with other adult age groups [14]. This is a concern because these age groups are a common time period to begin fatherhood [15,16,17].

It is well established that parents have a major influence on the dietary intake of young children, through their own behaviour, attitude, modelling, parenting styles and child feeding practices [18,19,20,21]. Until recently, the influence of fathers on the eating behaviours of children were overlooked, but a systematic review of 23 studies has shown father’s eating habits to be strongly related to a child’s dietary intake [22]. Despite this, fathers have rarely been engaged in family-based lifestyle interventions [23,24]. To our knowledge, only one lifestyle program has exclusively targeted fathers in parenting interventions to improve dietary behaviour. The ‘Healthy Dads, Healthy Kids’ program has evaluated dietary outcomes in efficacy [25], effectiveness [26] and dissemination trials [27]. The efficacy randomised controlled trial (RCT) [25] demonstrated favourable group-by-time reductions at 6 months in children’s energy intake (−1809 kJ/day, 95% CI: −3000, −619) and fathers’ portion size (−0.3, 95% CI: −0.5, −0.1) [25]. The effectiveness RCT [26] showed increases in children’s grain intake (*d* = 0.70), and reductions in fathers’ daily energy intake (*d* = 0.74), total sugars (*d* = 0.63), sodium (*d* = 0.58) and SSBs (*d* = 0.58) when compared with control at 3 months. Increases in fathers’ intakes of fruit (*d* = 0.71), vegetarian protein sources (*d* = 0.57) and percent energy from healthy, nutrient-dense, core foods (*d* = 0.86) were also evident. This study also found fathers’ eating patterns to be correlated with those of their children for several dietary variables [26]. In the non-randomised dissemination trial of ‘Healthy Dads, Healthy Kids’ in underserved communities, both children and fathers’ significantly increased vegetable intake at 3 months and 6 months (*p* < 0.05) and significantly reduced takeaway/fast food meals, sugar-sweetened beverages and snack intake at 3, 6 and 12 months (all *p* < 0.05) [27]. Despite the success, ‘Healthy Dads, Healthy Kids’ was conducted in children aged 5–12 years and in fathers with overweight or obesity, thus highlighting a gap in the literature for preschool-aged children and in fathers of a healthy weight.

Healthy Youngsters, Healthy Dads (HYHD) is the first healthy lifestyle program, internationally to exclusively target fathers and their preschool-aged children [28,29]. In a recent RCT, intervention fathers and preschool-aged children achieved statistically significant and clinically meaningful improvements in physical activity [29]. In children, findings were maintained at 9 months. The program also improved children’s sport skills competence and fathers parenting practices for physical activity and screen time [29]. In addition, a previous pre–post, single-arm feasibility trial of HYHD has been conducted in 24 father–preschool-aged children dyads and found medium effect sizes for children’s vegetable intake (*d* = 0.5), and fathers’ energy intake (*d* = 0.4) [28]. However, the impact of HYHD on dietary intake in a randomised controlled efficacy trial remains to be explored.

The aim of this paper was to evaluate the efficacy of the HYHD program on change in dietary intake in fathers and their preschool-aged children at the end of the intervention (10 weeks post-baseline). The secondary aim was to test whether any impact was sustained at long-term follow-up (9 months post-baseline). The final aim was to investigate whether an association exists in father–child dietary intakes.

## 2. Materials and Methods

### 2.1. Study Design

Full details of the Healthy Youngsters, Healthy Dads (HYHD) randomised controlled trial (RCT) have been reported elsewhere [29]. Briefly, family units (fathers and their preschool-aged child) were randomised in a 1:1 ratio to an 8 week intervention or a waitlist control group. The University of Newcastle Human Research Ethics Committee (H-2017-0381) approved this study, with fathers providing written informed consent and child assent before enrolment. This study was prospectively registered with the Australian New Zealand Clinical Trials Registry (ACTRN12619000105145) and the conduct of this study aligned with the CONSORT statement [30].

### 2.2. Participants

Between November 2018 and January 2019, families were recruited from the Newcastle region in New South Wales, Australia. The primary recruitment strategy included a University media release which featured in several local news outlets (newspaper, radio, television). Participants were also recruited via social media posts (Facebook, Instagram and Twitter), the distribution of flyers to local early childcare centres, and emails to participants of previous University programs. Eligible participants included a biological father, step-father, or male guardian of a child aged 3–5 years who lived with their child at least 50% of the week. Additionally, participants were required to pass a pre-exercise screening questionnaire for physical activity and available to attend assessment and program sessions. Fathers who indicated pre-existing health conditions were required to obtain doctor’s clearance prior to being accepted to the program. Children were eligible for the program if they were of preschool age (3–5 years), and not attending primary school (Kindergarten—Year 6) in the year of the trial.

### 2.3. The HYHD Intervention

HYHD was designed to motivate fathers to role model, encourage, and co-participate in fun and creative physical activity and healthy eating activities with their children, and vice versa. A full description of intervention components with associated behaviour change techniques and theoretical mediators targeted is detailed in Appendix A. Briefly, the 8 week HYHD program consisted of: two dads-only workshops (2 h each), eight consecutive weekly group sessions for fathers and children together (75 min each) and weekly home-based tasks (organized in an activity handbook). The dads-only workshops gave fathers evidenced-based strategies to optimise family diet quality and included topics such as information on Australian recommended daily food group intakes, planning meals, and support and dietary change strategies [29]. The workshops also included strategies to enhance fathers’ parenting practices to improve their children’s physical activity, dietary habits, social-emotional well-being and sports skills. The father–child weekly group sessions were split into two components: (i) A 20 min educational session based on a weekly theme (vegetables, fruit, water, etc.), with each theme linked to one of several, program animal characters (e.g., Reg Rhino—Vegetables) to engage the children, and (ii) a 55 min physical activity practical session. To provide motivation, children earned a weekly animal character sticker if they completed designated home tasks with their father, and a bonus sticker (e.g., banana) for completing more than one activity. An extensive overview of the HYHD intervention has been reported elsewhere [29].

### 2.4. Outcome Measures

Fathers’ self-reported dietary intake was assessed using the Australian Eating Survey (AES) food frequency questionnaires (FFQ) [31]. To reduce potential reporting bias for fathers reporting their child’s intake, each child’s mother completed the Toddler-specific version of the Australian Eating Survey (AES-T) [32]. Both the AES and AES-T are 120-item semi-quantitative food frequency questionnaires (FFQs) that have previously demonstrated reliability and comparative validity in adults and children aged 3.2 ± 0.5 years for assessing usual dietary intake [31,32,33]. In the current study, participants completed the online versions of the survey.

The AES assessed usual intake over the previous 10 weeks. When distributing the AES to fathers at post-program assessment, fathers were encouraged to “only consider the last 2 months (i.e., after you enrolled in the study).” The AES-T version was distributed to mothers using REDCap (Research Electronic Data Capture, Harris et al., 2009 [34]) with questions specific to the child’s intake (i.e., “How many pieces of fruit does your child eat?”). Usual food intake was assessed over the previous two months to align with the 8 week HYHD intervention.

The AES uses adult portion sizes, while the AES-T uses portion sizes for children aged five years and under derived from the 1995 National Nutrition Survey [35] and a standard portion size for foods with a ‘natural’ serving sizes (e.g., slice of bread). Response options were assessed on a Likert scale ranging from ‘never’ to ‘four or more times per day’ and for some drinks, up to ‘seven or more glasses per day’.

Nutrient intakes from the AES and AES-T were computed using data in the Australian food composition database, AusNut 1999 (All Foods) Revision 14 (Australian Government Publishing Service, 1995, Canberra, Australia [36]). The intake of macronutrients, core foods (including fruit, vegetables and dairy) and energy-dense, nutrient-poor (EDNP) foods (including takeaway/fast foods, confectionary and baked products) was expressed as a percentage of total energy intake.

Overall diet quality was determined using the Australian Recommended Food Score (ARFS), which is derived from the AES FFQ (fathers) or AES-T (children). ARFS uses a subset of 70 questions related to core, nutrient-dense foods recommended in the Australian Dietary Guidelines [37]. The ARFS is calculated by summing the points within eight sub-scales (vegetables, fruit, protein foods—meat/flesh, vegetarian sources of protein, grains, dairy, water and spreads/sauces). The total score ranges from zero to a maximum of 73 points, with a higher score indicative of a higher diet quality. The full scoring method for ARFS has been reported elsewhere [38]. ARFS has previously demonstrated validity in adults [39,40] and preschool-aged children [41].

Baseline demographic information and adiposity measures were also collected. This included participant age and socioeconomic status, determined using the Australian postal area index of relative socioeconomic advantage and disadvantage [42]. Body weight and height were measured using standardised procedures [28,29], both used the average of at least two measures with a tolerance of 0.1 kg for weight and 0.3 cm for eight for fathers and children. Measures were used to calculate BMI for fathers and BMI Z-score for children based on UK reference data [43] and LMS methods [44]. Body fat percentage for both fathers and their children were measured using the InBody720 bioelectrical impendence analyser, a multi-frequency bioimpedance device (Biospace Co., Ltd., Seoul, Korea).

#### 2.4.1. Sample Size

The primary study [29] had 80% power to detect a group-by-time difference of 1500-steps/day in children at post-intervention, which was the primary outcome of the trial. This calculation assumed a change score standard deviation of 2700 step/day (*p* < 0.05) and 85% retention rate. This study was not powered a priori to detect changes in the secondary outcomes including dietary intake. Multiplicity adjustments were not conducted for these secondary outcomes as they were intended to complement the primary outcome data and provide preliminary insights for definitive hypothesis testing in future studies.

#### 2.4.2. Statistical Analysis

All data analyses were conducted in SPSS Version 26 (IBM Corp., Armonk, NY, USA). All dietary variables were checked for accuracy and meeting the assumption of normality. Variables which had standardised scores above 3.29 were truncated to a value 1 unit greater than the next lowest value for that variable [45]. Intention-to-treat linear mixed models were used to assess all dietary outcomes for fathers and children for the impact of group (control or intervention), time (baseline, 10 weeks and 9 months) and the group-by-time interaction. Age and socioeconomic status were examined as potential covariates for both fathers and children; sex was also examined for children. All significant interactions and covariates were added to the final model. Statistical significance was set at *p* < 0.05 and effect sizes calculated using Cohen *d* (*d* = M1-M2/σ pooled). To interpret effect sizes, the following cutoffs were used: <0.3 small, 0.3 to 0.5 medium, >0.5 large [46]. The associations in father–child energy intake from macronutrients, core foods and EDNP foods were assessed using Pearson’s correlation test. Correlation strength was described as poor <0.20, moderate 0.2–0.6 and strong >0.6 [47].

Two sensitivity analyses were also conducted: (i) a completers’ analysis for participants who completed dietary assessment measures at all three time points (baseline, 10 weeks and 9 months) and (ii) a per-protocol analysis of HYHD intervention participants who complied well with the assigned treatment compared with the control group. ‘Per-protocol’ was defined in the clinical trials registry (ACTRN12619000105145) prior to commencing the trial as those that attended ≥75% of the sessions and completed ≥75% of the home-based tasks (measured by completing 4.5 or more of the 6 home tasks in the activity handbook on average each week).

## 3. Results

A total of 125 fathers and their children (*n* = 76, or 61% boys and *n* = 49 or 39% girls) participated in this study. Of these, dietary data were available for 119 (95%) fathers and 118 (94%) children at baseline. Subsequent dietary data were available for 91 (73%) fathers and 93 (74%) children at 10 weeks, and 90 (72%) fathers and 95 (76%) children at 9 months, respectively. Baseline characteristics for fathers and children are summarised in Table 1.

### 3.1. Baseline Dietary Intakes

#### 3.1.1. Children

Baseline dietary intakes of children in this current sample are generally representative when compared with preschool-aged children from national nutrition data [1]. At baseline, EDNP foods accounted for 28.9 ± 11.7% of total energy intake for children, which exceeds the Australian Dietary Guideline recommendation of no more than one serve for preschool-aged children (approximately 5–20% of daily energy intake [48]). The proportion of EDNP is comparable to national nutrition data (mean proportion of 30% for children aged 2–3 years, and 37% for those aged 4–8 years) [1]. The most commonly consumed type of EDNP foods were takeaway/fast foods, accounting for 6.6 ± 4.2% of total energy intake. The proportion of energy from saturated fat was 15.9 ± 2.9%, which exceeded the Acceptable Macronutrient Distribution Range (AMDR) of <10% for saturated fat [49]. Intakes are comparable to national data (proportion of energy intake from saturated fat is 14% in both early childhood age groups: 2–3 years and 4–8 years [7]). The mean energy intake of all children was 5840 ± 2533 kJ/day, which is equivalent to national nutrition data (mean of 5951 kJ/day for children aged 2–3 years, 7053 kJ/day for children aged 4–8 years) [1]. Children’s baseline dietary intake by group is provided in Table 2.

#### 3.1.2. Fathers

On the whole, intakes from fathers in this current sample are representative of the average Australian male in a comparable age category (aged 31–50 years) [7]. The pattern of fathers’ intake at baseline is comparable to their children’s intake, with EDNP foods substantially contributing to overall energy intake (38.6 ± 13.1%) and exceeding national recommendations of no more than 3 serves (approximately 5–20% of daily energy intake) for adult males aged 19–50 years [50]. The proportion of EDNP foods consumed in our sample is similar to national data (mean of 37% of total energy intake among males aged 31–50 years [7]). As was the case for children, takeaway/fast foods were the most consumed EDNP food among fathers (11.2 ± 7.1% of total energy intake). The proportion of energy from saturated fat among fathers (14.0 ± 2.7%) exceeded the AMDR for saturated fat (<10%) [49] and is comparable to national data (12% among males aged 31–50 years) [7]. Fathers’ baseline energy intake was 10,403 ± 2879 kJ/day, which is equivalent to the average intake from national data (10,220 kJ/day) [1]. Fathers’ baseline dietary intake by group is provided in Table 3.

### 3.2. Change in Dietary Intake

#### 3.2.1. Children

Table 2 shows the outcomes of intention-to-treat analyses of changes in children’s dietary intake at all time points. Briefly, at the end of the program, significant between- group effects, favouring the intervention were detected for the children’s increase in proportion of energy from healthy, nutrient-dense, core foods (+3.2%, *d* = 0.43). There were also significant group-by-time reductions in sodium intake (−182 mg/day, *d* = 0.38) and percentage energy from EDNP foods (−3.2%, *d* = 0.43) and prepacked snacks (−1.4%, *d* = 0.45). These significant findings were sustained at 9 months follow-up (all *p* < 0.05).

For some dietary variables, there were no significant group-by-time changes at the end of the program, but then significant findings were established at 9 months. These included significant reductions in energy intake (−1316 kJ/day, *d* = 0.46) and percentage energy from takeaway/fast foods (−1.4%, *d* = 0.41). There were also group-by-time effects, favouring the control group at 9 months for dietary fibre intake (+4.0 g/day, *d* = 0.42) and proportion of main meals without vegetables (−1.0, *d* = 0.37). For all other dietary variables, there were no significant group-by-time effects at any time point. The results were broadly consistent with those found in both the completers and per-protocol analyses (see Appendix A.

#### 3.2.2. Fathers

Table 3 shows the intention-to-treat analyses of change in fathers’ dietary intake at all time points. In summary, there were significant between-group effects favouring the intervention at the end of the program for total daily energy intake (−1372 kJ/day, *d* = 0.55), sodium intake (−350 mg/day, *d* = 0.64) and percentage energy from core foods (+4.8%, *d* = 0.49), EDNP foods (−4.8%, *d* = 0.49), and confectionary (−1.5%, *d* = 0.36). For all of these dietary variables, except sodium, the significant findings were sustained at 9 months follow-up (all *p* < 0.05).

For some dietary variables, there were no significant group-by-time changes at the end of the program, but then significant findings were established at 9 months. These included significant increases in diet quality score (+2.7, *d* = 0.37), percentage energy of grains (+4.8%, *d* = 0.61) and breakfast cereals (+1.9%, *d* = 0.46). For all other dietary variables in Table 3 there were no significant group-by-time effects at any time point. The results were broadly consistent with those produced in both the completers and per-protocol analyses (see Appendix A).

### 3.3. Correlation of Father–Child Dietary Intake at Baseline and for Change from Baseline

Table 4 reports the Pearson’s correlation coefficients for father–child dyads for percentage energy from macronutrients, healthy, nutrient-dense, core foods and EDNP foods at baseline and for the change from baseline to 10 weeks (post-intervention) and 9 months. At baseline, there were strong, significant positive correlations in father–child percent energy intake from takeaway/fast foods (r = 0.61, *p* < 0.001). There were also moderate, significant positive correlations for percent energy intake from core foods (r = 0.52, *p* < 0.001), EDNP foods (r = 0.52, *p* < 0.001), vegetarian protein sources (r = 0.42, *p* < 0.01), diet quality score (r = 0.40, *p* < 0.01), sugar-sweetened beverages (r = 0.35, *p* < 0.01), dairy (r = 0.31, *p* < 0.05), pre-packed snacks (r = 0.27, *p* < 0.05) and frequency of meals eaten with vegetables (r = 0.35, *p* < 0.01).

For change in intakes from baseline to 10 weeks, there were strong, significant positive correlations in father–child percent energy intakes from healthy, nutrient-dense, core foods (r = 0.62, *p* < 0.001) and EDNP foods (r = 0.62, *p* < 0.001). There were also moderate, significant positive correlations for percent energy from takeaway/fast foods (r = 0.52, *p* = 0.000), prepacked snacks (r = 0.40, *p* < 0.01), vegetarian protein sources (r = 0.39, *p* < 0.05), confectionary (r = 0.38, *p* < 0.05), and condiments (r = 0.36, *p* < 0.05).

For change in intakes from baseline to 9 months, there were moderate, significant positive correlations between change in fathers’ and child intakes for percent energy from breakfast cereals (r = 0.50, *p* < 0.01), sugar-sweetened beverages (r = 0.44, *p* < 0.01), vegetarian protein sources (r = 0.43, *p* < 0.01) and takeaway/fast foods (r = 0.39, *p* < 0.05).

## 4. Discussion

The aim of the current study was to (i) evaluate the efficacy of a family-based lifestyle intervention (‘Healthy Youngsters, Healthy Dads’) on change in dietary intake in fathers and their preschool-aged children at 10 weeks and after 9 months follow-up, compared to a waitlist control group and to (ii) investigate whether an association existed in father–child dietary intake. Findings indicate that the HYHD program resulted in medium to large intervention effects for some dietary variables in both fathers and young children when compared with controls. Specifically, at the end of the program (10 weeks), both fathers and children increased intakes of healthy, nutrient-dense core foods and reduced their intakes of EDNP foods and sodium. Children also reduced pre-packed snacks, and fathers also reduced energy intake and confectionary. At 9 months (6 months post-intervention), both fathers and children increased intakes of healthy, nutrient-dense core foods and reduced their energy intake and EDNP foods. Children also reduced intake of sodium and takeaway/fast foods, and fathers also increased overall diet quality score and intakes of grains and breakfast cereals and reduced intakes of confectionary. For some dietary variables in children (energy intake, takeaway/fast foods) and fathers (diet quality, grains, breakfast cereals), there were no significant group-by-time changes at the end of the program, but then significant findings were established at 9 months. This suggests that for these dietary variables, it may require longer periods of time for the changes to become acceptable and sustained. At 9 months, there were also increases in children’s intake of dietary fibre and reductions in the frequency of meals consumed without any vegetables that favoured the control group. There were no significant changes at any time point for either fathers or children for macronutrient intakes (protein, carbohydrate, fat or saturated fat), some core foods (vegetables, vegetarian sources of protein, fruit, meats, dairy), some EDNP foods (sugar-sweetened beverages, baked products, fatty meats and condiments), or meals with vegetables. There were moderate to strong associations in father–child dietary intakes at baseline and father–child dietary change scores for some of the dietary variables. This suggests that the dietary intake of fathers has an important influence on the dietary intakes of their children, and especially when it comes to addressing improvements in intake.

At baseline, the average dietary intakes of both fathers and children failed to meet national dietary recommendations for intakes of most foods groups and macronutrients. However, they were broadly representative of adult males and preschool-aged children in Australia [1,7]. In this context, the significant and sustained improvements in some dietary variables (e.g., healthy, nutrient-dense core foods and EDNP foods) among fathers and children in the intervention group are important findings, given the need for and paucity of successful dietary interventions targeting men [53] and preschool-aged children [54].

Limited comparisons can be made with the current literature due to the absence of interventions targeting fathers and preschool-aged children in the community and lack of comparable dietary data. Despite this, our findings are promising when compared with community interventions targeting father–child [25,26,27] and mother–preschool-aged child dyads [55]. Among children, our moderate post-intervention, group-by-time effect sizes were stronger than the ‘Healthy Dads, Healthy Kids’ effectiveness trial [26] over a similar time period for core foods (*d* = 0.43 vs. *d* = 0.20), EDNP foods (*d* = 0.43 vs. *d* = 0.20), pre-packed snacks (*d* = 0.45 vs. *d* = 0.00) and sodium (*d* = 0.38 vs. *d* = 0.35). In addition, mean between-group changes among children at 10 weeks were favourable when compared to a brief intervention with mother–preschool-aged child dyads (*Healthy Food to Kids*) [55] over a similar time period for core foods (+3.2% vs. −0.6%), and EDNP foods (−3.2% vs. +0.6%). This finding is notable, given that mother–preschool-aged child interventions are often limited by mothers completing questionnaires on behalf of their child. However, the child’s reported diet intake in HYHD was completed by the non-intervention target (mother-proxy) to reduce reporting bias.

The small group-by-time effect sizes for children’s change in fruit (10 weeks: *d* = 0.05, 9 months: *d* = 0.16) and vegetables intake (10 weeks: *d* = 0.23, 9 months: *d* = 0.08) are comparable to a recent systematic review which evaluated the impact of nutrition interventions on children’s fruit and vegetable intake [56]. There were three preschool/school studies identified in this review that incorporated a parent component and used an FFQ to assess dietary intake [57,58,59]. In these studies, effects sizes for change in fruit intake were *d* = 0.10 [58], *d* = 0.14 [57], and *d* = 0.19 [59], while change in vegetable intake was: *d* = 0.08 [59], *d* = 0.10 [58], and *d* = 0.13 [57]. The limited effects may be attributed to child eating behaviour traits such as ‘food fussiness’ and food neophobia (i.e., avoidance of new foods) which peaks in children aged 2–5 years [60]. A recent systematic review and meta-analysis identified ‘repeated taste exposure’ as the best strategy to increase vegetable intake in young children but a minimum of 8–10 exposures is required [54]. The young children in HYHD and other parent-based interventions may not have had enough exposure to vegetables during short-term intervention periods. In addition, it has been shown that men are often resistant to improving vegetable intake [61], which is reflective of the fathers in this current study. The minimal changes among fathers may have ultimately influenced the small effects in children.

At 10 weeks, our moderate effect sizes among fathers are comparable to another intervention targeting father–child dyads (*‘Healthy Dads, Healthy Kids’* [26]) over a similar time period for sodium (*d* = 0.64 vs. *d* = 0.58), and confectionary (*d* = 0.36 vs. *d* = 0.35). However, the effect sizes were weaker for nutrient-dense core foods (*d* = 0.49 vs. *d* = 0.86), EDNP foods (*d* = 0.49 vs. *d* = 0.86) and energy intake (*d* = 0.55 vs. *d* = 0.74). The eligibility criteria in ‘Healthy Dads, Healthy Kids’ required fathers to have a BMI greater than 25 kg/m^2^ with a greater focus placed on weight loss and energy restriction. This may explain the weaker effects in HYHD, especially as a recent systematic review on men’s weight loss programs found that 89% of weight loss programs that provided a specific energy restriction target (e.g., *eat X kJ per day, or reduce usual intake by X kJ per day*) were effective, compared to 46% that did not do this [62]. Despite this, the significant and sustained reductions in total energy intake in HYHD are notable given the sample included fathers within a healthy BMI range and the major focus was not weight loss or energy restriction. The reductions in energy intake are likely due to respective reductions in EDNP foods and confectionary.

High sodium intake is the leading global dietary risk factor for non-communicable disease mortality and morbidity, accounting for 3 million deaths and 70 million Disability Adjusted Life Years (DALYs) in 2017 [63]. Given this, the group-by-time, post-intervention reductions in sodium intake among HYHD fathers are especially important, as these reductions take the daily intakes below the national recommended limit for adult males of 2000 mg/day [51]. However, added salt was not measured so intakes were likely higher. It is likely that these reductions in sodium are due to fathers replacing EDNP foods with healthy, nutrient-dense core foods. Additionally, the group-by-time, post-intervention reductions in sodium intakes among HYHD children (−182 mg/day) are encouraging because excess sodium can begin to adversely impact on blood pressure from birth [64], while high blood pressure can track into adulthood and translate to higher rates of cardiovascular disease [65] in adults. Going forward, greater focus should be placed on achieving further reductions that are sustained in the long term.

The positive findings in HYHD may be due to the program’s focus on limiting coercive control (e.g., not using food as a reward), improving meal structure (e.g., family mealtimes) and enhanced autonomy (e.g., involving children in meal preparation), which are considered the three most important influences in the development of child feeding practices [66]. Additionally, the program engaged and involved children through: program animal characters (e.g., Reg Rhino—Vegetables), enabling children to select home-tasks (e.g., *have a competition with dad to see who can get the most number of different coloured vegetables on their fork?*) and rewarding children with weekly animal character stickers for completing designated home tasks with their father. In addition, the program educated fathers on using an authoritative parenting style (e.g., a combination of high parental control and positive stimuli to the child’s autonomy, including nurturing/warmth, rational communication and receptiveness [67]) as this has been shown to improve the amount and type of limit setting and reinforcement for healthy food choices in children [68,69]. Finally, the program sought to improve the modelling and involvement of fathers in supporting implementation of healthy eating at home, with a particular focus on supporting mothers’ who commonly are the solo focus in this endeavor [70]. The positive role modelling from fathers supports the emerging literature regarding fathers’ influence on children’s lifestyle behaviours [20,71,72,73]. Specifically, the interactions during mealtimes between fathers and children have been shown to positively and negatively influence children’s long-term eating behaviour [20]. These findings should be viewed in conjunction with the knowledge that maternal parenting practices also play a key role in children’s dietary intake [71,74]. Thus, the influences of both parents are vital. However, fathers are hugely under-represented in family-based lifestyle programs, especially in programs for younger children [23,75]. Therefore, it is imperative that family-based lifestyle programs consider engaging both fathers and mothers.

The null findings for some dietary variables (e.g., protein, fats) align with similar interventions [26,55]. However, the lack of study power is likely to have contributed to null findings for some diet outcomes, as t study was only powered for the primary outcome of the program (steps/day). Despite this, the findings of this efficacy study will help to provide preliminary insights for definitive hypothesis testing in future larger-scale effectiveness studies. Furthermore, potential inconsistencies in message framing (e.g., loss-frame for decreasing EDNP and gain-frame for nutrient-rich core foods) could also provide an explanation of a lack of behaviour change for some dietary outcomes. Future interventions may need to include a more intensive nutrition focus to optimise dietary patterns of fathers and their children (e.g., greater support when at home, more time allocated at sessions and a greater involvement of mothers at some sessions) and greater consistency in messaging with gain-framed messages shown to be more effective in influencing children’s healthy food choices [76].

The moderate to strong associations in father–child dietary intake at baseline for some foods (e.g., diet quality score, core foods, dairy, EDNP foods, SSB, pre-packed snacks, takeaway/fast foods and meals with vegetables) support the importance of paternal modelling on a child’s dietary intake. This is congruent with a recent systematic review and meta-analysis of 18 studies, that confirmed parental modelling as one of the most influential parenting practices in their child’s food consumption [19]. However, only one of the studies in this review solely focused on fathers. To date, there have been no systematic reviews or meta-analyses conducted to explore associations between fathers and their child’s dietary intakes. Two cross-sectional studies have explored dietary correlations between fathers and their preschool-aged children [77,78]. The study by Walsh et al. [77] found positive associations with fruit, sweet snacks and takeaway/fast foods but no associations for savoury snacks and vegetables, while the study by Vollmer et al. [78] reported positive associations for the father’s and child’s overall diet quality (β = 0.39; *p* < 0.0001). The current study also found several significant father–child changes in intake at 10 weeks (vegetarian protein sources, core foods, EDNP foods, pre-packed snacks, takeaway/fast foods and condiments) and at 9 months (vegetarian sources of protein, breakfast cereal, SSB and takeaway/fast foods). To date, only two studies (‘Healthy Dads, Healthy Kids pilot RCT and community RCT’ [25,26]) investigated the relationship between the father–child changes in dietary intake and found significant associations for grains [25], fruit, carbohydrates, vegetarian sources of protein and meals consumed with vegetables [26]. Overall, this indicates that fathers play an important role influencing the dietary intakes of their young children. However, in order to reinforce these findings, a more meaningful representation of fathers is required in future research.

A possible justification for the positive associations between fathers and their children was the emphasis on paternal modelling within HYHD. Specifically, fathers were educated about diet quality and how to model and influence their children’s dietary behaviours. Positive interactions were then stimulated through encouraging fathers to involve children in purchasing, preparing and eating meals. This offers an ideal setting to promote observational learning, whereby children perceive their parents eating behaviours as the ‘norm’ and mimic their parents eating behaviours [20,79].

Strengths of this study include the use of two validated FFQs that used the same questions for fathers and children so questions were directly comparable. Additionally, mothers were used as a proxy for children’s intakes to reduce reporting bias and allow comparison with the literature, given the scarcity of research using fathers to report child dietary intakes. There are limitations associated with using self-report FFQs as they provide approximations of usual intake and can be associated with mis-reporting bias. However, the FFQs used in this study have been validated against several objective dietary biomarkers (doubly labelled water [32], red blood cell membrane fatty acids [80] and plasma carotenoids [33]). In addition, to address energy intake misreporting, food group intakes were expressed as a percentage of total energy intake. The use of automated self-administered 24 h dietary recalls (e.g., ASA-24) may have provided a more accurate measure of diet [81]. However, this method was not used due to a lack of comparative data among preschool-aged children, greater researcher and participant burden and variations in administration and reporting of 24 h recalls (e.g., number of weekends and weekdays, forgotten food approach) [82]. Instead, an FFQ that has been previously validated in preschool-aged children using doubly labelled water [32], with additional validation studies in adults was used [31]. Comparable data for preschool-aged children and father–child population groups using the FFQ were also available and hence it was chosen for this current study [25,26,55]. While previous research has demonstrated validity of using bioelectrical impedance analysis for use in preschool-aged children [83], there have been no studies that have validated the model used in this current study (InBody 720) in preschool-aged children. As such, further validation studies are required in this group. Finally, the analyses were not powered to detect changes in dietary intake. Therefore, results should be interpreted with caution and require replication in larger-scale effectiveness studies that are powered to detect dietary changes.

## 5. Conclusions

The HYHD program targeted fathers as agents of change for their pre-school-aged children, and vice versa. This study demonstrated some medium to large, group-by-time effect sizes for change in dietary intake of fathers and young children following the intervention, compared to waitlist controls. There were also significant associations in father–child dietary intake at baseline and change in intakes at 10 weeks and 9 months for some of the dietary variables. Although further research is required, the current study provides preliminary support for targeting fathers to act as positive role models to improve the dietary intakes of their preschool-aged children.

## Figures and Tables

**Table 1 nutrients-13-03306-t001:** Demographic characteristics of study participants.

Children	Control (*n* = 64)		HYHD (*n* = 61)		Total (*n* = 125)	
	Mean	SD	Mean	SD	Mean	SD
Age (year) (*n* = 125)	3.9	0.5	4.0	0.5	3.9	0.5
Weight (kg) (*n* = 124)	17.2	2.3	17.7	2.4	17.5	2.4
Height (cm) (*n* = 124)	103.3	6.3	104.0	5.5	103.6	5.9
Body fat mass (%) (*n* = 121)	17.6	5.7	17.8	7.7	17.7	6.7
BMI (kg/m^2^) (*n* = 124)	16.1	1.1	16.3	1.4	16.2	1.3
BMI z-score ^a^ (*n* = 124)	0.2	0.8	0.4	1.0	0.3	0.9
	** *n* **	**%**	** *n* **	**%**	** *n* **	**%**
Male	42	65.6%	34	55.7%	76	60.8%
**Fathers**	**Control (*n* = 64)**		**HYHD (*n* = 61)**		**Total (*n* = 125)**	
	**Mean**	**SD**	**Mean**	**SD**	**Mean**	**SD**
Age (year) (*n* = 125)	38.4	4.9	37.6	5.9	38.0	5.4
Weight (kg) (*n* = 125)	90.9	17.3	90.9	19.5	90.9	18.3
Height (cm) (*n* = 125)	179.5	7.5	179.6	7.3	179.6	7.4
Body fat mass (%) (*n* = 124)	23.1	8.5	22.3	8.3	22.7	8.3
BMI (kg/m^2^) (*n* = 125)	28.2	4.8	28.1	5.1	28.1	4.9
	** *n* **	**%**	** *n* **	**%**	** *n* **	**%**
Relationship status (*n* = 125)						
*Single*	0	0.0%	2	3.3%	2	1.6%
*Married/de facto*	63	98.4%	59	96.7%	122	97.6%
*Separated*	1	1.6%	0	0.0%	1	0.8%
Socioeconomic status ^b^ (*n* = 125)						
*1 (lowest)*	1	1.6%	1	1.6%	2	1.6%
*2*	16	25.0%	18	29.5%	34	27.2%
*3*	26	40.6%	22	36.1%	48	38.4%
*4*	16	25.0%	18	29.5%	34	27.2%
*5 (highest)*	5	7.8%	2	3.3%	7	5.6%

^a^ BMI-z calculated using the LMS method (World Health Organization growth reference centiles) [44]. ^b^ Socioeconomic status by population quintile for SEIFA Index of Relative Socioeconomic Advantage and Disadvantage [42]. HYHD; Healthy Youngsters, Healthy Dads.

**Table 2 nutrients-13-03306-t002:** Change in dietary intake of children enrolled in the Healthy Youngsters, Healthy Dads (HYHD) RCT (intention to treat).

			Baseline	10 Week Change from Baseline (Mean, 95% CI)			9 Month Change from Baseline (Mean, 95% CI)		
Outcome	NRV [48,49,51,52]	Group	Mean (SE)	Within Group ^a^	Mean Difference between Groups ^b^	*p*-Value (Cohen’s *d*)	Within Group ^c^	Mean Difference between Groups ^b^	*p*-Value (Cohen’s *d*)
Energy (kJ/day) *^,d,e^	-	Intervention	5940 (314)	−414 (−1150, 322)	−679 (−1715, 357)	0.198 (0.24)	+561 (−159, 1282)	**−1316 (−2346, −287)**	**0.012 (0.46)**
		Control	5752 (308)	+265 (−464, 994)			**+1877 (1143, 2612)**		
Diet Quality Score (max score is 73, higher score = better diet quality)									
Total ARFS	-	Intervention	31.6 (1.2)	+0.8 (−0.7, 2.3)	0.0 (−2.1, 2.1)	0.999 (0.00)	+0.4 (−1.1, 1.8)	0.0 (−2.0, 2.1)	0.975 (0.01)
		Control	34.2 (1.2)	+0.8 (−0.7, 2.2)			+0.4 (−1.1, 1,9)		
Macronutrients (% of total energy intake or g/day)									
Protein (%)	AMDR: 15–25%	Intervention	16.2 (0.3)	+0.3 (−0.2, 0.8)	+0.4 (−0.3, 1.1)	0.226 (0.22)	+0.4 (−0.1, 0.9)	+0.5 (−2, 1.2)	0.129 (0.28)
		Control	16.7 (0.3)	−0.1 (−0.6, 0.4)			−0.1 (−0.6, 0.4)		
Carbohydrate (%)	AMDR: 45–65%	Intervention	48.2 (0.7)	−0.3 (−1.3, 0.8)	−1.4 (−2.9, 0.1)	0.071 (0.33)	+0.0 (−1.2, 1.3)	−1.4 (−3.1, 0.4)	0.117 (0.29)
		Control	46.9 (0.7)	**+1.1 (0.0, 2.1)**			**+1.4 (0.2, 2.7)**		
Total sugars (g/day) ^d^	-	Intervention	92.7 (5.2)	−8.7 (−19.2, 1.8)	−14.2 (−29.0, 0.6)	0.061 (0.35)	+4.1 (−6.1, 14.4)	−14.6 (−29.3, 0.1)	0.052 (0.36)
		Control	85.5 (5.1)	+5.5 (−4.9, 15.9)			**+18.7 (8.2, 29.2)**		
Fat (%)	AMDR: 20–35%	Intervention	36.0 (0.6)	−0.2 (−1.1, 0.6)	+0.9 (−0.3, 2.2)	0.141 (0.27)	−0.7 (−1.8, 0.4)	+0.9 (−0.7, 2.4)	0.268 (0.20)
		Control	36.9 (0.6)	−1.2 (−2.1, −0.3)			−1.6 (−2.7, −0.5)		
Saturated fat (%)	<10%	Intervention	15.6 (0.3)	−0.2 (−0.9, 0.4)	+0.5 (−0.4, 1.5)	0.245 (0.21)	−0.7 (−1.3, 0.0)	+0.4 (−0.6, 1.3)	0.467 (0.13)
		Control	16.1 (0.3)	**−0.8 (−1.4, −0.1)**			**−1.0 (−1.7, −0.4)**		
Micronutrients									
Fibre (g/day) ^d^	AI: 14 g (aged 3 years), 18 g (aged 4–5 years)	Intervention	17.7 (1.2)	−0.7 (−3.2, 1.8)	−2.3 (−5.8, 1.2)	0.205 (0.23)	**+2.7 (0.3, 5.1)**	**−4.0 (−7.5, −0.5)**	**0.024 (0.42)**
		Control	17.3 (1.1)	+1.6 (−0.9, 4.0)			**+6.7 (4.2, 9.2)**		
Sodium (mg/day) ^d^	UL: 1000 mg (aged 3 years), 1400 mg (aged 4–5 years)	Intervention	1302 (70)	−121 (−246, 4)	**−182 (−357, −6)**	**0.043 (0.38)**	+117 (−58, 292)	**−382 (−631, −132)**	**0.003 (0.56)**
		Control	1236 (69)	+61 (−63, 184)			**+499 (321, 677)**		
Healthy, nutrient-dense core foods (% of total energy intake)									
Core foods ^f^ (%)	≥85%	Intervention	68.9 (1.5)	**+4.0 (2.1, 5.9)**	**+3.2 (0.5, 6.0)**	**0.021 (0.43)**	**+2.2 (0.3, 4.1)**	**+3.5 (0.8, 6.2)**	**0.013 (0.46)**
		Control	73.1 (1.4)	+0.8 (−1.1, 2.7)			−1.2 (−3.2, 0.7)		
Vegetables ^f^ (%)	-	Intervention	6.0 (0.6)	+0.7 (−0.3, 1.6)	+0.8 (−0.5, 2.1)	0.204 (0.23)	+0.1 (−0.8, 1.0)	+0.3 (−1.0, 1.6)	0.661 (0.08)
		Control	7.3 (0.6)	−0.1 (−1.1, 0.8)			−0.2 (−1.1, 0.7)		
Fruit (%)	-	Intervention	10.4 (0.7)	**+1.4 (0.2, 2.7)**	−0.2 (−2.0, 1.5)	0.781 (0.05)	+1.1 (−0.1, 2.3)	+0.7 (−1.0, 2.5)	0.397 (0.16)
		Control	9.9 (0.7)	**+1.7 (0.5, 2.9)**			+0.3 (−0.9, 1.6)		
Meats ^d^ (%)	-	Intervention	7.5 (0.6)	+0.7 (−0.5, 1.9)	+0.3 (−1.3, 2.0)	0.687 (0.07)	+0.9 (−0.2, 2.1)	+0.7 (−0.9, 2.4)	0.390 (0.16)
		Control	7.7 (0.6)	+0.4 (−0.8, 1.6)			+0.2 (−1.0, 1.4)		
Vegetarian protein sources (%)	-	Intervention	2.7 (0.4)	+0.1 (−0.5, 0.7)	+0.0 (−0.8, 0.9)	0.940 (0.01)	+0.1 (−0.5, 0.7)	+0.1 (−0.8, 0.9)	0.912 (0.02)
		Control	3.1 (0.4)	+0.1 (−0.5, 0.7)			+0.05 (−0.5, 0.6)		
Grains ^d^ (%)	-	Intervention	19.1 (0.9)	+0.9 (−1.0, 2.8)	+0.3 (−2.4, 3.0)	0.849 (0.03)	**+2.0 (0.2, 3.9)**	−0.3 (−3.0, 2.3)	0.802 (0.05)
		Control	19.9 (0.9)	+0.6 (−1.3, 2.5)			**+2.4 (0.5, 4.3)**		
Breakfast cereals (%)	-	Intervention	9.4 (0.7)	+0.3 (−1.2, 1.8)	+0.6 (−1.5, 2.7)	0.585 (0.10)	+0.4 (−1.0, 1.9)	+1.0 (−1.1, 3.1)	0.364 (0.17)
		Control	9.3 (0.7)	−0.3 (−1.8, 1.2)			−0.5 (−2.0, 1.0)		
Dairy ^d^ (%)	-	Intervention	25.6 (1.3)	+0.2 (−2.4, 2.8)	+2.2 (−1.5, 5.8)	0.239 (0.22)	−2.1 (−4.7, 0.4)	+1.8 (−1.8, 5.4)	0.322 (0.18)
		Control	27.4 (1.2)	−2.0 (−4.5, 0.6)			−4.0 (−6.5, −1.4)		
Energy-dense nutrient-poor (EDNP) foods (% of total energy intake)									
EDNP foods ^f^ (%)	<1 serve (~5–20%)	Intervention	31.0 (1.5)	**−4.0 (−6.0, −2.1)**	**−3.2 (−6.0, −0.5)**	**0.021 (0.43)**	**−2.2 (−4.1, −0.3)**	**−3.5 (−6.2, −0.8)**	**0.013 (0.46)**
		Control	26.9 (1.4)	−0.8 (−2.7, 1.1)			+1.2 (−0.7, 3.2)		
Sugar-sweetened beverages (%)	-	Intervention	0.9 (0.2)	−0.2 (−0.6, 0.2)	−0.2 (−0.8, 0.3)	0.357 (0.17)	−0.2 (−0.5, 0.2)	−0.1 (−0.6, 0.3)	0.551 (0.11)
		Control	0.7 (0.2)	+0.1 (−0.3, 0.4)			−0.0 (−0.4, 0.3)		
Prepacked snacks (%)	-	Intervention	4.4 (0.5)	**−1.6 (−2.4, −0.8)**	**−1.4 (−2.5, −0.3)**	**0.015 (0.45)**	**−1.6 (−2.4, −0.9)**	**−2.2 (−3.3, −1.1)**	**0.000 (0.72)**
		Control	3.0 (0.5)	−0.2 (−1.0, 0.6)			+0.5 (−0.2, 1.3)		
Confectionary (%)	-	Intervention	5.0 (0.5)	−1.0 (−1.8, −0.2)	−0.6 (−1.7, 0.6)	0.331 (0.18)	−0.4 (−1.2, 0.4)	+0.2 (−0.9, 1.4)	0.682 (0.08)
		Control	4.8 (5.7)	−0.4 (−1.3, 0.4)			−0.6 (−1.4, 0.2)		
Baked products (%)	-	Intervention	5.6 (0.4)	−0.9 (−1.7, −0.1)	−0.7 (−1.8, 0.5)	0.267 (0.20)	−0.4 (−1.2, 0.4)	−0.5 (−1.6, 0.7)	0.407 (0.15)
		Control	5.1 (0.4)	−0.2 (−1.1, 0.6)			+0.1 (−0.8, 0.9)		
Takeaway/fast foods ^d,f^ (%)	-	Intervention	7.3 (0.5)	−0.4 (−1.3, 0.4)	−0.8 (−2.0, 0.4)	0.180 (0.25)	**+0.5 (−0.4, 1.3)**	**−1.4 (−2.5, −0.2)**	**0.025 (0.41)**
		Control	5.8 (0.5)	+0.4 (−0.5, 1.2)			+1.8 (1.0, 2.7)		
Condiments (%)	-	Intervention	3.3 (0.3)	+0.03 (−0.6, 0.7)	−0.01 (−0.9, 0.9)	0.990 (0.00)	−0.3 (−0.9, 0.3)	−0.03 (−0.9, 0.8)	0.940 (0.01)
		Control	3.1 (0.3)	+0.04 (−0.6, 0.7)			−0.3 (−0.9, 0.4)		
Fatty meats (%)	-	Intervention	1.9 (0.2)	+0.0 (−0.4, 0.4)	+0.0 (−0.5, 0.6)	0.933 (0.02)	+0.1 (−0.3, 0.5)	+0.2 (−0.4, 0.7)	0.600 (0.10)
		Control	1.8 (0.2)	−0.0 (−0.4, 0.4)			−0.0 (−0.4, 0.4)		
Meals (% of total energy intake)									
Meals with vegetables (%) ^g^	-	Intervention	4.8 (0.5)	+0.6 (−0.3, 1.6)	+0.2 (−1.1, 1.6)	0.741 (0.06)	−0.1 (−1.0, 0.9)	−0.3 (−1.7, 1.0)	0.650 (0.08)
		Control	4.7 (0.5)	+0.4 (−0.6, 1.4)			+0.2 (−0.7, 1.2)		
Meals without vegetables (%) ^h^	-	Intervention	1.6 (0.3)	−0.1 (−0.7, 0.4)	+0.1 (−0.7, 0.8)	0.874 (0.03)	+0.7 (−0.0, 1.4)	**+1.0 (0.0, 2.1)**	**0.047 (0.37)**
		Control	1.6 (0.3)	−0.2 (−0.7, 0.3)			−0.3 (−1.1, 0.4)		

Bold denotes a significant difference (*p* < 0.05). * 1 kcal = 4.186 kJ. ^a^ 10 week value minus baseline; ^b^ Within-group difference (intervention) minus within-group difference (control); ^c^ 9 month value minus baseline; ^d^ Adjusted for child’s age; ^e^ Truncated to account for outliers [45] (>3.29 SD truncated to next highest value plus 1); ^f^ Adjusted for child’s sex; ^g^ Composite score of all protein meals that were consumed with vegetables, ^h^ Composite score of all protein meals that were consumed without vegetables. ARFS, Australian Recommended Food Score; NRV, Nutrient Reference Values; AMDR, Acceptable Macronutrient Distribution Ranges; AI, Acceptable Intake; UL, upper limit.

**Table 3 nutrients-13-03306-t003:** Change in dietary intake of fathers enrolled in the Healthy Youngsters, Healthy Dads (HYHD) RCT (intention to treat).

			Baseline	10 Week Change from Baseline (Mean, 95% CI)			9 Month Change from Baseline (Mean, 95% CI)		
Outcome	NRV [49,50,51,52]	Group	Mean (SE)	Within Group ^a^	Mean Difference between Groups ^b^	*p*-Value (Cohen’s *d*)	Within Group ^c^	Mean Difference between Groups	*p*-Value (Cohen’s *d*)
Energy (kJ day) *^d,e^	-	Intervention	10,515 (412)	**−1092 (−1717, −467)**	**−1372 (−2272, −473)**	**0.003 (0.55)**	−649 (−1304, 6)	**−1189 (−2095, −282)**	**0.010 (0.47)**
		Control	10,325 (404)	+252 (−395, 898)			+505 (−120, 1130)		
Diet Quality Score (max score is 73, higher score = better diet quality)									
Total ARFS	-	Intervention	31.8 (1.2)	+1.1 (−0.7, 2.9)	+1.4 (−1.2, 4.1)	0.279 (0.20)	+0.9 (−1.0, 2.8)	**+2.7 (0.1, 5.3)**	**0.05 (0.37)**
		Control	34.5 (1.1)	−0.4 (−2.2, 1.5)			−1.8 (−3.6, 0.0)		
Macronutrients (% of total energy intake)									
Protein (%)	AMDR: 15–25%	Intervention	17.0 (0.4)	+0.8 (0.0, 1.6)	+0.7 (−0.4, 1.8)	0.203 (0.23)	+0.8 (−0.0, 1.6)	+0.8 (−0.4, 1.9)	0.183 (0.24)
		Control	17.1 (0.4)	+0.1 (−0.7, 0.9)			+0.0 (−0.7, 0.8)		
Carbohydrate (%)	AMDR: 45–65%	Intervention	43.4 (0.9)	+0.4 (−1.0, 1.9)	+0.3 (−1.8, 2.4)	0.809 (0.04)	−0.1 (−1.6, 1.4)	+0.7 (−1.4, 2.8)	0.622 (0.11)
		Control	44.1 (0.9)	+0.2 (−1.3, 1.7)			−0.8 (−2.2, 0.7)		
Fat (%)	AMDR: 20–35%	Intervention	36.1 (0.7)	−0.5 (−1.7, 0.7)	−0.3 (−2.0, 1.4)	0.737 (0.06)	−0.4 (−1.6, 0.9)	−0.4 (−2.2, 1.3)	0.617 (0.09)
		Control	35.9 (0.7)	−0.2 (−1.5, 1.0)			+0.1 (−1.1, 1.3)		
Saturated fat (%)	<10%	Intervention	13.5 (0.3)	−0.4 (−1.1, 0.2)	−0.5 (−1.4, 0.5)	0.310 (0.19)	−0.3 (−1.0, 0.4)	−0.8 (−1.7, 0.1)	0.093 (0.31)
		Control	13.3 (0.3)	+0.0 (−0.6, 0.7)			+0.5 (−0.2, 1.1)		
Alcohol (%)	-	Intervention	3.8 (0.5)	−0.9 (−1.4, 0.3)	−0.7 (−1.5, 0.0)	0.060 (0.35)	−0.2 (−1.5, 1.0)	−1.0 (−2.8, 0.7)	0.248 (0.21)
		Control	2.8 (0.5)	−0.1 (−0.7, 0.4)			+0.8 (−0.4, 2.0)		
Micronutrients									
Fibre (g/day) ^d^	AI: 30 g/day (male aged 19–70 years)	Intervention	30.7 (1.3)	−0.0 (−1.8, 1.8)	−1.2 (−3.8, 1.4)	0.362 (0.17)	+0.1 (−3.5, 3.8)	−1.7 (−6.7, 3.3)	0.503 (0.16)
		Control	32.4 (1.3)	+1.2 (−0.7, 3.1)			+1.8 (−1.6, 5.3)		
Sodium (mg/day)	SDT: <2000 mg/day (male aged 19–70 years).	Intervention	2271 (97)	**−321 (−459, −184)**	**−350 (−549, −151)**	**0.001 (0.64)**	−189 (−627, 250)	−455 (−1062, 152)	0.140 (0.27)
		Control	2230 (95)	+29 (−115, 173)			+266 (−153, 686)		
Core foods (% of total energy intake)									
Core foods	≥85%	Intervention	59.8 (1.6)	**+5.9 (3.4, 8.3)**	**4.8 (1.3, 8.3)**	**0.008 (0.49)**	**+4.4 (1.8, 6.9)**	**+4.2 (0.7, 7.7)**	**0.020 (0.43)**
		Control	63.0 (1.6)	+1.1 (−1.4, 3.6)			+0.2 (−2.3, 2.6)		
Vegetables (%)	-	Intervention	8.1 (0.6)	+1.2 (0.3, 2.1)	+0.7 (−0.6, 2.1)	0.273 (0.20)	+0.9 (−0.1, 1.9)	+0.4 (−0.9, 1.8)	0.518 (0.12)
		Control	8.7 (0.6)	+0.5 (−0.5, 1.4)			+0.5 (−0.5, 1.4)		
Fruit (%)	-	Intervention	5.9 (0.6)	**+1.6 (0.5, 2.6)**	+1.4 (−0.0, 2.9)	0.057 (0.35)	+0.3 (−0.8, 1.4)	+0.5 (−1.0, 2.0)	0.487 (0.13)
		Control	6.6 (0.5)	+0.1 (−0.9, 1.2)			−0.3 (−1.3, 0.8)		
Meats (%) ^f^	-	Intervention	13.6 (0.9)	+1.3 (−0.6, 3.2)	+1.5 (−1.2, 4.2)	0.277 (0.20)	+0.7 (−1.2, 2.7)	+1.1 (−1.6, 3.9)	0.410 (0.15)
		Control	13.5 (0.9)	−0.2 (−2.1, 1.7)			−0.4 (−2.3, 1.5)		
Vegetarian protein sources (%) ^f^	-	Intervention	3.7 (0.4)	−0.0 (−0.7, 0.6)	+0.2 (−0.8, 1.2)	0.680 (0.08)	−0.4 (−1.1, 0.3)	−0.5 (−1.5, 0.4)	0.288 (0.19)
		Control	3.6 (0.4)	−0.2 (−0.9, 0.5)			+0.1 (−0.6, 0.8)		
Grains (%)	-	Intervention	19.4 (1.1)	+1.2 (−0.8, 3.1)	+0.8 (−2.0, 3.6)	0.582 (0.10)	+3.0 (0.9, 5.0)	**4.8 (1.9, 7.6)**	**0.001 (0.61)**
		Control	21.6 (1.1)	+0.4 (−1.6, 2.4)			−1.8 (−3.7, 0.2)		
Breakfast cereals (%)	-	Intervention	5.8 (0.7)	+0.6 (−0.1, 1.7)	+0.3 (−1.2, 1.9)	0.649 (0.08)	+0.9 (−0.2, 2.0)	**1.9 (0.4, 3.5)**	**0.013 (0.46)**
		Control	7.0 (0.7)	+0.3 (−0.8, 1.4)			−1.0 (−2.1, 0.0)		
Dairy (%) ^d^	-	Intervention	8.9 (0.8)	+0.3 (−1.4, 2.1)	−0.4 (−2.9, 2.1)	0.745 (0.06)	−0.0 (−1.9, 1.8)	−1.7 (−4.3, 0.8)	0.180 (0.25)
		Control	9.2 (0.8)	+0.7 (−1.1, 2.6)			+1.7 (−0.1, 3.4)		
Energy-dense nutrient-poor (EDNP) foods (% of total energy intake)									
EDNP foods (%)	<3 serves (~5–20%)	Intervention	40.2 (1.6)	**−5.9 (−8.3, −3.4)**	**−4.8 (−8.3, −1.3)**	**0.008 (0.49)**	**−4.4 (−6.9, −1.8)**	**−4.2 (−7.7, −0.7)**	**0.020 (0.43)**
		Control	37.0 (1.6)	−1.1 (−3.6, 1.4)			−0.2 (−2.6, 2.3)		
Sugar-sweetened beverages (%)	-	Intervention	2.5 (0.5)	+−0.2 (−0.9, 0.6)	−0.2 (−1.3, 0.9)	0.713 (0.07)	−0.9 (−1.5, −0.3)	−0.4 (−1.2, 0.5)	0.403 (0.15)
		Control	1.7 (0.5)	+0.0 (−0.7, 0.8)			−0.5 (−1.1, 0.1)		
Prepacked snacks (%)	-	Intervention	3.6 (0.4)	−0.6 (−1.3, 0.1)	−0.8 (−1.8, 0.3)	0.141 (0.23)	−0.2 (−0.9, 0.6)	−0.6 (−1.7, 0.4)	0.216 (0.23)
		Control	2.5 (0.3)	+0.2 (−0.5, 0.9)			+0.5 (−0.2, 1.2)		
Confectionary (%)	-	Intervention	6.4 (0.6)	**−2.0 (−3.0, −0.9)**	**−1.5 (−3.0, −0.0)**	**0.047 (0.36)**	**−1.9 (−3.0, −0.8)**	**−1.8 (−3.3, −0.3)**	**0.022 (0.42)**
		Control	5.8 (0.6)	−0.5 (−1.6, 0.6)			−0.1 (−1.1, 1.0)		
Baked products (%)	-	Intervention	4.3 (0.5)	+0.0 (−0.8, 0.9)	−0.3 (−1.5, 1.0)	0.655 (0.08)	+0.0 (−0.9, 0.9)	−0.3 (−1.5, 1.0)	0.667 (0.08)
		Control	4.4 (0.5)	+0.3 (−0.6, 1.2)			+0.3 (−0.6, 1.1)		
Takeaway/fast foods (%)	-	Intervention	11.1 (0.9)	−1.1 (−2.4, 0.2)	−0.0 (−1.9, 1.8)	0.956 (0.01)	−0.9 (−2.1, 0.3)	−0.5 (−2.2, 1.1)	0.531 (0.11)
		Control	11.3 (0.9)	−1.0 (−2.4, 0.3)			−0.4 (−1.5, 0.8)		
Condiments (%)	-	Intervention	3.1 (0.4)	−0.5 (−1.1, 0.2)	−0.6 (−1.6, 0.4)	0.247 (0.21)	−0.4 (−1.1, 0.3)	−0.4 (−1.4, 0.6)	0.441 (0.14)
		Control	3.4 (0.4)	0.1 (−0.6, 0.8)			−0.0 (−0.7, 0.7)		
Fatty meats (%)	-	Intervention	2.7 (0.2)	−0.3 (−0.8, 0.1)	−0.2 (−0.9, 0.4)	0.451 (0.14)	−0.1 (−0.5, 0.4)	+0.5 (−0.2, 1.1)	0.139 (0.27)
		Control	2.7 (0.2)	−0.1 (−0.5, 0.4)			−0.6 (−1.0, −0.1)		
Meals *(% of total energy intake)*									
Meals with vegetables (%) ^g^	-	Intervention	8.5 (0.6)	+0.8 (−0.5, 2.2)	+1.0 (−0.9, 3.0)	0.297 (0.19)	+0.7 (−0.7, 2.1)	+1.3 (−0.7, 3.2)	0.198 (0.24)
		Control	8.7 (0.6)	−0.2 (−1.6, 1.2)			−0.6 (−1.9, 0.8)		
Meals without vegetables (%) ^h^	-	Intervention	2.7 (0.5)	−0.0 (−1.0, 1.0)	−0.1 (−1.6, 1.3)	0.843 (0.04)	−0.4 (−1.3, 0.6)	−0.5 (−1.8, 0.8)	0.477 (0.13)
		Control	2.5 (0.5)	+0.1 (−0.9, 1.1)			+0.1 (−0.8, 1.0)		

Bold denotes a significant difference. * 1 kcal = 4.186 kJ. ^a^ 10 week value minus baseline; ^b^ Within-group difference (intervention) minus within-group difference (control); ^c^ 9 month value minus baseline; ^d^ Adjusted for SES; ^e^ Truncated to account for outliers [45] (>3.29 SD truncated to next highest value plus 1); ^f^ Adjusted for father’s age; ^g^ Composite score of all protein meals that were consumed with vegetables; ^h^ Composite score of all protein meals that were consumed without vegetables. ARFS, Australian Recommended Food Score; NRV, Nutrient Reference Value; AMDR, Acceptable Macronutrient Distribution Ranges; AI, Acceptable Intake; SDT, Suggested Dietary Target.

**Table 4 nutrients-13-03306-t004:** Baseline and change score correlations for dietary variables of fathers and their children participating in the ‘Healthy Youngsters, Healthy Dads’ RCT.

Father-Child Dietary Variables	Baseline Correlations		10 Week Change Score Correlations		9 Month Change Score Correlations	
	r	*p*-Value	r	*p*-Value	r	*p*-Value
Energy (kJ day) (%)	−0.10	0.442	0.14	0.374	0.26	0.107
Diet Quality Score (ARFS)	**0.40**	**0.002**	0.13	0.396	0.20	0.223
Protein (%)	0.20	0.130	−0.07	0.638	0.05	0.773
Carbohydrate (%)	0.13	0.333	−0.15	0.344	−0.06	0.730
Fat (%)	−0.04	0.751	−0.10	0.545	0.02	0.925
Saturated fat (%)	−0.05	0.709	−0.12	0.427	0.16	0.332
Fibre (g/day)	0.03	0.817	0.23	0.139	0.21	0.195
Sodium (mg/day)	−0.02	0.867	0.07	0.673	0.28	0.077
Healthy, nutrient-dense core foods (%)	**0.52**	**0.000**	**0.62**	**0.000**	0.25	0.118
Vegetables (%)	0.04	0.750	0.16	0.320	0.09	0.567
Fruit (%)	0.10	0.476	0.12	0.446	0.20	0.209
Meats (%)	0.25	0.068	−0.12	0.453	0.06	0.692
Vegetarian protein sources (%)	**0.42**	**0.001**	**0.39**	**0.010**	**0.43**	**0.005**
Grains (%)	−0.04	0.771	0.05	0.770	0.25	0.117
Breakfast cereals (%)	0.10	0.476	0.25	0.103	**0.50**	**0.001**
Dairy (%)	**0.31**	**0.019**	0.17	0.273	0.21	0.195
EDNP foods (%)	**0.52**	**0.000**	**0.62**	**0.000**	0.25	0.118
Sugar-sweetened beverages (%)	**0.35**	**0.008**	0.23	0.143	**0.44**	**0.004**
Prepacked snacks (%)	**0.27**	**0.040**	**0.40**	**0.008**	0.18	0.274
Confectionary (%)	0.24	0.073	**0.38**	**0.012**	−0.04	0.817
Baked products (%)	−0.00	0.980	0.13	0.420	0.13	0.433
Takeaway/fast foods (%)	**0.61**	**0.000**	**0.52**	**0.000**	**0.39**	**0.011**
Condiments (%)	0.00	0.982	**0.36**	**0.017**	0.19	0.226
Fatty meats (%)	0.00	0.973	0.04	0.808	0.05	0.762
Meals with vegetables (%)	**0.35**	**0.007**	0.12	0.452	0.13	0.414
Meals without vegetables (%)	0.25	0.065	0.26	0.088	0.03	0.878

EDNP, energy dense nutrient poor. ARFS, Australian Recommended Food Score. Data in bold highlight statistically significant correlations.

## Data Availability

Data available on request due to restrictions. The data presented in this study are available on request from the corresponding author. The data are not publicly available due to ethical reasons.

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
