# Peer review of "Dietary Outcomes of the ‘Healthy Youngsters, Healthy Dads’ Randomised Controlled Trial"

_nutrients, 2021, doi:10.3390/nu13103306_

Round 1

Reviewer 1 Report

Thank you for the opportunity to review this timely and important manuscript regarding the role of fathers in their pre-school children’s weight-related behaviors. The primary aim of this paper was to determine the efficacy of a 10-wk family-based lifestyle intervention on change in dietary intake between fathers and their participating children. Additionally, the authors sought to explore associations between father-child dietary intakes. Fathers’ parenting practices and behaviors are vastly understudied within the context of family-based pediatric research and this leaves gaping holes in understanding etiology of childhood conditions, including pediatric obesity. This research is incredibly valuable for informing future intervention work with families. This paper is well-written. However, there were a few concerns as indicated below.

Methods:

Has the InBody720 been validated for use in pre-school aged children? (Line 188)

Considering the referenced paper for expanding upon study design is under review, it would benefit the reader and future research with fathers to provide additional information regarding recruitment. Were preschools targeted for recruiting fathers and children? Pediatric offices (e.g., pediatrician, dentist)? Or were mass mailings used to recruit? (Lines 115-123)

Were fathers questioned regarding their role of fathers in coordinating food/meals for their children, including percentage of time spent preparing foods, grocery shopping, mealtimes? What percentage of fathers were married or living with their children’s mother? The lack of research conducted among fathers is concerning, and while the authors did note this within the discussion as a reason for focusing on fathers exclusively (Lines 451-456), it may be difficult to differentiate individual differences between mothers and fathers, especially if fathers are married/living as married.

Methods/Discussion:

The authors used FFQs for measuring dietary intake. Considering 24-hour dietary recalls are a better measure of self-reported intake and there is availability of the ASA-24 for Australia, addressing why these were not used is important. (Lines 497-509)

Discussion:

Explanations for lack of behavior change may not be limited to time to become acceptable and sustained but also the possibility of different messaging (loss-frame vs gain-frame) that be conducive to increasing or reducing consumption for certain foods compared to others. For example, messaging for increasing consumption of healthy foods (fruits, vegetables) may be different from decreasing consumption of takeaway foods. (Discussion, Lines 353-354)

It is understood that the goal of the intervention was not to reduce caloric intake explicitly. However, reductions in EDNP and confectionary foods could ultimately lead to reductions in caloric intake, especially if these foods are replaced by healthier alternatives. (Discussion, Lines 412-418)

Author Response

Dear reviewer,

Thank you for your constructive critique of the manuscript. We believe the suggested changes have greatly improved the quality of the manuscript.

Each comment has been considered by the authors, and we have addressed in turn. The manuscript has been revised to incorporate appropriate changes and we have indicated where the revised text is now located in the manuscript.

Please see full response in attached document. 

Reviewer 2 Report

Thank you for the invitation to review this interesting article.

A large percentage of overweight and obese in both adults and children population indicates a gap between the dietary recommendations and usual dietary intake. Hence, it is desirable to create effective programs aimed at spreading the principles of healthy nutrition.

The idea and the organization of The HYHD study efficiently draw attention. I was curious about the details and effects of this intervention focused on the father-child relationship and the conclusions are promising.

The article is well-written and gives a clear picture of the actual state of knowledge.

I have only 2 small comments/suggestions:

Line 186 – I think the Authors probably meant height, not weight

And the second one is page numbering - it requires correction

Author Response

(The authors gave the same response as above.)
